# MicroRNA-34 Family in Cancers: Role, Mechanism, and Therapeutic Potential

**DOI:** 10.3390/cancers15194723

**Published:** 2023-09-26

**Authors:** Junjiang Fu, Saber Imani, Mei-Yi Wu, Ray-Chang Wu

**Affiliations:** 1Key Laboratory of Epigenetics and Oncology, The Research Center for Preclinical Medicine, Southwest Medical University, Luzhou 646000, China; 2Shulan International Medical College, Zhejiang Shuren University, Hangzhou 310022, China; 3Department of Biochemistry and Molecular Biology, School of Medicine, University of Maryland Baltimore, Baltimore, MD 21201, USA; 4Department of Biochemistry and Molecular Medicine, The George Washington University, Washington, DC 20052, USA

**Keywords:** microRNA, miR-34, tumor suppressor, miRNA therapy

## Abstract

**Simple Summary:**

The dysregulation of the miR-34 family has been shown to play an important role in tumorigenesis and cancer progression. In this review, we summarized the current knowledge on the functions of the miR-34 family in human cancers, its emerging role as a diagnostic and prognostic biomarker, and its potential as a novel therapeutic in cancer.

**Abstract:**

MicroRNA (miRNA) are small noncoding RNAs that play vital roles in post-transcriptional gene regulation by inhibiting mRNA translation or promoting mRNA degradation. The dysregulation of miRNA has been implicated in numerous human diseases, including cancers. miR-34 family members (miR-34s), including miR-34a, miR-34b, and miR-34c, have emerged as the most extensively studied tumor-suppressive miRNAs. In this comprehensive review, we aim to provide an overview of the major signaling pathways and gene networks regulated by miR-34s in various cancers and highlight the critical tumor suppressor role of miR-34s. Furthermore, we will discuss the potential of using miR-34 mimics as a novel therapeutic approach against cancer, while also addressing the challenges associated with their development and delivery. It is anticipated that gaining a deeper understanding of the functions and mechanisms of miR-34s in cancer will greatly contribute to the development of effective miR-34-based cancer therapeutics.

## 1. Introduction

MicroRNAs (miRNAs) are short RNA molecules, around 21–23 nucleotides long, that control gene expression by binding to target mRNA’s 3′-untranslated regions (3′-UTRs) primarily through a seed region from nucleotides 2 to 8 [1]. They are involved in diverse biological processes, and their dysfunction is linked to diseases such as cancer, neurodegenerative disorders, and heart conditions [2]. In cancer, miRNAs regulate critical processes like tumor formation, cell proliferation, angiogenesis, invasion, and metastasis by impinging on the expression of key oncogenes and tumor suppressor genes. Depending on the targets, miRNAs can act as either oncogenes or tumor suppressors, emphasizing their complex roles in cancer development.

Among the tumor-suppressive miRNAs, the miR-34 family (miR-34s) has emerged as a prominent player in various human diseases, particularly cancer. In recent years, there has been increasing interest in unraveling the intricate relationship between miR-34s and cancer [3,4,5]. This comprehensive review aims to provide an in-depth analysis of the role of miR-34 family members in cancer. We will discuss the epigenetic mechanisms through which miR-34s modulate cancer cell proliferation, apoptosis, invasion, and metastasis. Additionally, we will explore the potential of miR-34 family members as markers for cancer diagnosis and prognosis, as well as their prospects as therapeutics in cancer treatment.

## 2. Biosynthesis of miR-34s and Regulation of miR-34 Expression

### 2.1. Biosynthesis of miR-34s

The miR-34 family comprises three members, miR-34a, miR-34b, and miR-34c, encoded by genes located on chromosomes 1 and 11 [3]. MiR-34b and miR-34c share a common primary seed sequence on chromosome 11, while miR-34a is encoded on chromosome 1. Comparative analysis shows that miR-34a shares 86% identity with miR-34b and 82% with miR-34c [3,6,7]. miR-34a and miR-34c also share identical seed sequences with the miR-449 superfamily, including miR-449a, miR-449b, and miR-449c. miR-34a typically exhibits higher expression levels than miR-34b and miR-34c in most human cells [8].

The biosynthesis of miR-34s is a multistep process that occurs in the nucleus and cytoplasm. RNA polymerase II or III transcribes miR-34 genes in the nucleus, generating a long hairpin-shaped molecule known as pri-miRNA. The DROSHA endonuclease cleaves the pri-miRNA into the 80–100-nucleotide-long pre-miRNAs within the nucleus. Exportin-5 transports the pre-miR-34s to the cytoplasm, where the DICER endonuclease further processes them into double-stranded mature miR-34s, approximately 20–23 nucleotides long [9]. These mature miRNAs associate with Argonaute proteins to form the RNA-induced silencing complex (RISC). One strand becomes the mature miRNA, while the other is degraded. miRNA-mediated gene silencing depends on the level of complementarity between the miR-34 seed sequence and target mRNA binding sites. Full complementarity may lead to mRNA degradation, while partial complementarity inhibits its translation [6,10,11].

The alignment of mature miR-34a, miR-34b, and miR-34c sequences is depicted in Figure 1. It is worth mentioning that the differential expression levels of miR-34s indicate their distinct and specific functions in the regulation of gene expression. The precise mechanisms through which miRNAs exert their effects on gene expression are still under investigation, and ongoing research endeavors are expanding our knowledge of the intricate and dynamic roles played by miRNAs in cellular physiology and development of diseases.

### 2.2. Regulation of miR-34 Expression

The regulation of miR-34 expression involves multiple mechanisms that contribute to its dysregulation in cancer. Transcriptional regulation by various factors plays a crucial role in controlling miR-34 levels. The tumor suppressor protein p53 directly binds to the miR-34 gene promoter, activating its transcription. Similarly, other transcription factors, such as Elk-1, STAT3, Snail, Slug, ZEB1, and ZEB2, have been implicated in the regulation of miR-34 [6].

The dysregulation of EMT-TFs contributes to the acquisition of mesenchymal characteristics by cancer cells, resulting in their increased invasiveness and migratory capacity. EMT-TFs recruit chromatin-modifying enzymes, such as histone deacetylases and DNA methyltransferases (DNMTs), to the promoter regions of miR-34 genes. This recruitment leads to histone deacetylation and DNA methylation, causing a repressive chromatin state and inhibition of the miR-34 transcription [6].

Epigenetic modifications, including CpG island methylation and histone deacetylation, further contribute to the dysregulation of miR-34 in cancers. The methylation of CpG islands in the promoter regions of miR-34 genes results in gene silencing, while histone deacetylation leads to a more compact chromatin structure that hinders transcriptional activity [4]. These epigenetic modifications are commonly found in various cancer types, such as lung [12,13], multiple myeloma [14], kidney [15,16], gastric [17], breast [18,19], colorectal [20], hepatocellular [21], prostate [22], and ovarian [23,24] cancers.

Understanding the intricate regulatory networks involving transcriptional factors, EMT-TFs, and epigenetic modifications provides valuable insights into the molecular processes underlying cancer progression. Targeted therapies aimed at restoring miR-34 expression and inhibiting EMT-associated metastasis may hold promise for future therapeutic interventions in cancer treatment.

Given the tumor-suppressive function of miR-34s, correlations between the expression of miR-34s and patient survival were interrogated using a Cox proportional hazards regression and a Kaplan–Meier survival analysis (https://kmplot.com/analysis/index.php?p=service&cancer=pancancer_mirna) (access date: 24 April 2023) [25]. In multiple distinct cancer types, the high expression of miR-34s, such as miR-34a (Figure 2B, *p* = 0.0021), miR-34b (Figure 2F, *p* = 0.0001), and miR-34c, is correlated with longer survival of patients with liver hepatocellular carcinoma (HCC). For example, Chen et al. reported that patients with a lower level of serum exosomal miR-34a showed worse overall survival than patients with its high expression, implying it is a noninvasive marker for the diagnosis and prognosis of HCC [26].

## 3. The Roles of miR-34s in Cancers

Given the global health impact of cancer and its status as a leading cause of death, understanding the roles of miR-34s in cancer is crucial for the development of effective therapies [4,5,27]. The functions of miR-34s in cancer are diverse and involve the modulation of critical signaling pathways and processes. Extensive research has identified over 700 genes that are either predicted or validated targets of miR-34s [28]. These genes are involved in critical signaling pathways implicated in cancer progression, including the MAPK, Notch, Wnt, PI3K/AKT, p53, and Ras pathways. Additionally, miR-34s target processes critical for cancer migration, invasion, and EMT [28,29,30,31]. Studies have demonstrated that a systemic delivery of miR-34a, combined with chemotherapy or radiotherapy, can effectively inhibit tumor progression [32,33].

Because of the large number of target genes, miR-34s are able to regulate a wide array of cellular functions. For example, miR-34s inhibit cell proliferation and survival by targeting the MAPK pathway in various cancer cells. Similarly, miR-34s regulate the Notch and Wnt signaling pathways, which are crucial for cell fate determination, differentiation, and maintenance of stem cells. The dysregulation of these pathways is commonly observed in cancer, and miR-34s can restore normal signaling by targeting key genes within these pathways [28].

Moreover, the PI3K/AKT pathway, which regulates cell survival and growth, is another target of miR-34s [34,35]. By inhibiting genes within the PI3K/AKT pathway, miR-34s can suppress cancer cell proliferation and induce cell death. In the realm of tumor suppression, miR-34a acts as a tumor suppressor that selectively targets the p53 protein, famously known as the “guardian of the genome”. The intricate interplay between p53 and miR-34 establishes positive feedback loops that facilitate tumor suppression. Specifically, activation of miR-34a enhances the functional capabilities of p53, resulting in critical cellular processes such as cell cycle arrest, DNA repair, and apoptosis [36]. The miR-34 family assumes a crucial role in mediating the functions of p53, thereby significantly contributing to the preservation of genomic stability [37,38,39]. miR-34a exhibits specific targeting of proapoptotic molecules such as SIRT1 and FasR, which are key players in p53/miR-34-induced apoptosis and in the regulation of apoptotic signaling [18,31,40,41,42,43]. Moreover, the miR-34 family interacts with a diverse array of oncogenic regulators, including TP53, NOTCH1, and SMAD4, as well as components of the Wnt and TGF-β signaling pathways [29,30,31]. These interactions form a complex regulatory network that influences cellular apoptosis and modulates crucial signaling cascades, adding to our understanding of the multifaceted roles played by the miR-34 family in cellular processes.

Understanding the regulatory mechanisms of miR-34s and their targets is essential for the development of effective cancer therapies [28]. Continued research into the precise functions and regulatory networks of miR-34s in different types of cancer will provide valuable insights for the design of targeted treatments and personalized medicine approaches [5]. By harnessing the potential of miR-34s, we can advance the fight against cancer and improve patient outcomes. Table 1 summarizes the current miR-34 targets.

In the subsequent section, we will provide a concise overview of the latest progress in understanding the specific functions of miR-34s in various types of cancer.

### 3.1. The Role of miR-34s in Breast Cancer

Breast cancer is a significant cause of mortality in women and the most commonly diagnosed cancer in the United States, with an estimated 297,790 new cases and 43,170 expected deaths in 2023 [27]. In breast cancer, the expression of the miR-34 family is downregulated compared to normal tissues. Specifically, high expression of miR-34a has been correlated with less aggressive cancer behavior, although it has not been correlated with patient survival [90].

miR-34a plays a crucial role in maintaining mammary epithelium homeostasis. miR-34a expression is observed during luminal commitment and differentiation, and it inhibits the expansion of mammary gland stem cells and early progenitor cells through the suppression of Wnt/beta-catenin signaling. Interestingly, the function of miR-34a in stem/progenitor cells is conserved in human breast cancer. When miR-34a is expressed in triple-negative mesenchymal-like cells, which are enriched in cancer stem cells (CSCs), it leads to a luminal-like differentiation, restriction of the cancer stem cell pool, and inhibition of tumorigenesis [91]. In addition, it has been shown that miR-34a regulates cell survival and therapy resistance in breast cancer by targeting HDAC1 and HDAC7 in breast cancer [64]. These findings highlight the potential of miR-34a in breast cancer treatment.

Triple-negative breast cancer (TNBC) is the most aggressive subtype of breast cancer and is correlated with the lowest overall survival rates. New therapies for TNBC are urgently needed for this deadly disease. The inhibition of programmed cell death-ligand 1 (PD-L1), an immune checkpoint, has emerged as a promising treatment strategy. It has been shown that the tumor suppressor protein p53 can downregulate PD-L1 expression through miR-34a, thereby inhibiting the growth of TNBC cells [78]. Circulating miR-34 family expression levels could serve as noninvasive prognostic biomarkers in TNBC patients. Notably, low expression of circulating miR-34c has been correlated with poor prognosis in TNBC [92]. Recent research on single-nucleotide polymorphism (SNP) in miRNA and target sites has shed new light on the mechanistic contributions of miRNAs in carcinogenesis and their impact on cancer susceptibility and development. SNPs in miRNA and miRNA-binding sites of protein-coding genes further contribute to the complex and diverse roles of miRNAs in gene expression regulation. For instance, the rs4938723 C > T polymorphism in the promoter region of miR-34b/c impacts GATA-X binding and subsequently alters miR-34b/c expression [93]. Tsiakou et al. reported that the rs4938723 T > C polymorphism is correlated with an unfavorable prognosis in TNBC patients [94]. Several recent studies also demonstrated that the rs4938723 C > T polymorphism is correlated with a higher risk of hepatocellular [21,95] and prostate [96] cancers.

### 3.2. The Role of miR-34s in Lung Cancer

Lung cancer accounts for a significant number of cancer-related deaths worldwide. In the United States, the year 2023 is expected to witness a staggering 235,760 newly diagnosed cases of lung cancer, with a sobering estimate of approximately 131,880 lives being lost to this formidable disease [27]. MicroRNAs also play a crucial role in regulating vital pathways associated with the development of lung tumors, underscoring their potential as promising targets for innovative therapeutic interventions in the context of this particularly lethal form of cancer [22,97]. Study by Li et al. [79] revealed that the long noncoding RNA (lncRNA) X-inactive specific transcript (XIST) upregulates the expression of PD-L1 by targeting miR-34a-5p. By influencing the viability, apoptosis, migration, invasion, and cytokine secretion of lung cancer cells, this regulatory mechanism ultimately facilitates the upregulation of immunosuppressive molecules. Consequently, it contributes to the development of an immunesuppressive tumor microenvironment, which can impede effective anticancer immune responses and potentially influence disease progression in lung cancer [79]. Interestingly, miR-34a and miR-34b/c appear to have distinct functions in lung cancer cells [98]. Kim et al. [85] revealed that miR-34b/c, and not miR-34a, enhance cell attachment and suppress cell growth and invasion. miR-34a and miR-34b/c are capable of blocking lung metastasis in a syngeneic mouse model. Compared to miR-34a, the expression of miR-34b/c also leads to a greater reduction in the expression of mesenchymal markers such as Cdh2 and Fn1, while increasing the expression of epithelial markers including Cldn3, Dsp, and miR-200. Additionally, miR-34c has been shown to reduce the malignant characteristics of lung cancer cells via targeting TBL1XR1/Wnt/beta-catenin signaling [85].

The expression of miR-34b-3p has been reported to be downregulated in non-small-cell lung cancer (NSCLC) [99]. Feng et al. [54] reported that miR-34b suppresses proliferation and cell cycle progression and induces apoptosis in NSCLC cells by targeting CDK4 [54]. Furthermore, low levels of circulating miR-34 family members have been correlated with poorer prognoses in NSCLC patients, suggesting that miR-34a/c can serve as novel prognostic biomarkers for NSCLC [13]. Knockout of miR-34a/b/c has been revealed to promote mutant KRAS-driven lung tumor progression in vivo. Significantly lower expression levels of miR-34s have been found in metastatic lung adenocarcinoma samples compared to nonmetastatic samples. Hypermethylation of miR-34s promoters has predominantly been observed in lung adenocarcinoma, suggesting that miR-34 methylation could be utilized as a prognostic marker for NSCLC patients [100,101]. Additionally, the inhibition of miR-34a expression by lncRNA MCM3AP-AS1 was reported to enhance cell invasion, migration, and tumor formation in NSCLC [102].

Competing endogenous RNA (ceRNA) is a novel concept that involves various RNA molecules, including mRNA, circular RNA, pseudogene transcripts, and lncRNA. These molecules regulate each other’s expression by acting as sponges for microRNAs [103]. LncRNA MCM3AP-AS1 has been reported to enhance cell invasion, migration, and tumor formation in NSCLC by epigenetically inhibiting miR-34a [102]. Through this mechanism, MCM3AP-AS1 acts as a ceRNA, sequestering miR-34a and preventing its interaction with its target genes. The interruption of miR-34a with its target genes contributes to the altered behavior of cancer cells, promoting their invasive and migratory abilities. Another lncRNA, RP11-805J14.5, regulates CCND2, a gene involved in cell cycle regulation, by acting as a ceRNA/sponge for miR-34 and miR-139 in lung adenocarcinoma [53]. By binding to these microRNAs, RP11-805J14.5 effectively reduces their availability for binding to CCND2 mRNA, thereby increasing the expression of CCND2 and potentially promoting cell cycle progression in lung cancer cells.

The ceRNA mechanism provides a complex network of interactions among different RNA molecules, allowing them to communicate and influence gene expression patterns. In lung cancer, the dysregulation of ceRNA networks involving miR-34a, MCM3AP-AS1, RP11-805J14.5, and other associated molecules can have significant implications for the development and progression of the disease. Future studies on these ceRNA networks will provide valuable insights into the underlying mechanisms of lung cancer and potentially uncover novel therapeutic targets.

### 3.3. miR-34s in Hepatocellular Carcinoma

Hepatocellular carcinoma (HCC) poses a significant health challenge due to its high incidence and mortality rates. In 2023, the United States is expected to witness approximately 42,230 new cases of HCC, with a tragic toll of around 30,230 anticipated deaths [27]. When delving into the intricate landscape of miR-34s in HCC, it becomes evident that these microRNAs play a substantial role in governing crucial molecular pathways associated with the development and progression of hepatocellular carcinoma [104]. Researchers are actively exploring the therapeutic potential of harnessing miR-34s to combat this formidable liver cancer, opening new avenues for innovative treatment strategies [105]. Hepatocellular carcinoma (HCC) is a prevalent form of liver cancer and one of the leading causes of death worldwide. The dysregulation of miR-34s has been implicated in the development and progression of HCC. The miR-34a-c-MYC-CHK1/CHK2 axis has been identified as an important regulatory pathway in HCC. It was reported that this axis inhibits cancer stem cell-like functions and enhances radiosensitivity in HCC. By targeting MYC and CHK1/CHK2, miR-34s can suppress the stemness properties of cancer cells and increase their sensitivity to radiation therapy, potentially improving treatment outcomes [74].

A study by Jiao et al. [106] demonstrated that miR-34s are downregulated in tumor tissues compared to the matched noncancerous tissues. The downregulation of miR-34s in HCC is associated with poor prognosis. This suggests that the reduced expression of miR-34s may contribute to the aggressive nature of HCC and serve as a valuable prognostic marker for the disease [106]. Furthermore, the correlation between polymorphisms in the promoter region of pri-miR-34b/c and the risk of HCC has been studied. The results revealed that certain genetic variations in this region are associated with increased susceptibility to HCC, highlighting the potential role of miR-34s in the genetic predisposition to HCC [107].

In addition to tissue samples, the levels of miR-34a in serum exosomes from HCC patients have also been evaluated as a potential biomarker. Exosomal miR-34a has been revealed to be significantly downregulated in HCC patients, suggesting its potential utility as a noninvasive diagnostic and prognostic biomarker for HCC [26]. Overall, the dysregulation of miR-34s is implicated in hepatocellular carcinoma, with their reduced expression associated with poor prognosis and increased cancer stem cell-like functions. It is expected that future studies on miR-34s and the identification of their downstream targets may contribute to the development of novel therapeutic strategies and biomarkers for HCC.

### 3.4. The Role of miR-34s in Head and Neck Cancer

The downregulation of miR-34a has been implicated in the development of head and neck squamous-cell carcinoma (HNSCC). The downregulation of miR-34a promotes cancer progression by upregulating the mesenchymal–epithelial transition (MET) oncogene and modulating tumor immune evasion mechanisms [108]. Furthermore, miR-34a is involved in regulating Flotillin-2 and the MEK/ERK1/2 pathway, which are important signaling pathways in HNSCC [59].

Nasopharyngeal carcinoma (NPC) is a rare malignancy that poses a significant health threat, particularly in countries like China. Various risk factors, including the Epstein–Barr virus, contribute to the development of NPC. Recent studies have shed light on the role of circular RNAs (circRNAs) in NPC pathogenesis. CircRNAs are a unique class of single-stranded RNAs with closed-loop structures, generated through a process called back-splicing from pre-mRNA [60]. CircCRIM1, a circRNA, has been identified to play a key role in NPC. It acts as a ceRNA by sequestering miR-34c, thereby promoting NPC progression through the miR-34c/FOSL1 pathway [60]. This dysregulation of miR-34c and its downstream targets contributes to the tumorigenicity and aggressiveness of NPC. These findings highlight the significance of miR-34s in head and neck cancer, specifically HNSCC and NPC. Understanding the role of miR-34s and their regulatory mechanisms in these cancers can provide valuable insights into the underlying molecular pathways and potential therapeutic targets.

### 3.5. The Role of miR-34s in Esophageal Squamous-Cell Carcinoma

Esophageal squamous-cell carcinoma (ESCC) is a highly aggressive and lethal form of cancer. A downregulation of miR-34a expression has been revealed in ESCC tissues compared to normal tissues, and its expression levels have been found to correlate with the clinicopathological characteristics of ESCC [109]. A decreased expression of miR-34a and p53 has been reported in ESCC tissues, and an overexpression of miR-34a has been shown to inhibit ESCC cell migration and invasion by directly suppressing the expressions of MMP-2, MMP-9, and FNDC3B [71]. miR-34a exerts its inhibitory effects on ESCC migration and invasion through the regulation of different target genes, including Yin Yang-1 [89], MMP9, MMP14 [110], E2F5 [58], and FOXM1 [61]. Furthermore, the hypermethylation of miR-34b/c has been correlated with early clinical stages and tumor differentiation in Kazakh ESCC patients [111]. These findings highlight the crucial role of miR-34s in ESCC, where miR-34a dysregulation and its downstream effects fuel ESCC aggressiveness, promoting migration, invasion, and tumor progression. Understanding the intricate mechanisms involving miR-34s in ESCC can pave the way for novel diagnostic and therapeutic approaches in combating this deadly disease.

### 3.6. The Role of miR-34s in Gastric Cancer

Gastric cancer (GC) is a prevalent malignancy affecting the digestive system. The dysregulation of miR-34s in GC tissues is commonly observed and can be attributed to various factors, including p53 mutations, miR-34 promoter DNA methylation, and histone modifications. These alterations contribute to the downregulation of miR-34 expression in GC.

As a tumor suppressor gene, miR-34 exerts important regulatory functions in GC by inducing apoptosis, cell senescence, and cell cycle arrest, thereby inhibiting uncontrolled cell proliferation. MiR-34s also play a role in suppressing cell migration and metastasis, which are crucial steps in the spread of cancer. These effects are achieved through the regulation of several different target genes and signaling pathways [5,112]. For example, miR-34c has been implicated in GC progression by targeting MAP2K1, a gene involved in the MAPK pathway. By downregulating MAP2K1 expression, miR-34c inhibits the activation of the MAPK pathway, which is correlated with cell proliferation, migration, and invasion in GC cells. This suggests that the restoration of miR-34c expression or the modulation of its target genes could potentially serve as therapeutic strategies for GC treatment. Furthermore, the long noncoding RNA linc01106 has been identified as a key regulator in GC through its interaction with miR-34a. Linc01106 acts as a sponge for miR-34a, thus sequestering miR-34a and preventing its binding to target mRNAs [68]. Silencing linc01106 has been shown to suppress the malignant behaviors of GC cells by restoring the inhibitory effects of miR-34a on its downstream target gene MYCN. This indicates that targeting the linc01106/miR-34a/MYCN pathway might offer potential therapeutic benefits for GC patients [73]. The dysregulation of miR-34s in GC underscores their critical role in the development and progression of this malignancy. Understanding the molecular mechanisms underlying miR-34-mediated regulation can provide valuable insights into the biology of GC and facilitate the identification of novel therapeutic targets and diagnostic markers to improve patient outcomes.

### 3.7. The Role of miR-34s in Colon Carcinoma

Colon carcinoma is a significant contributor to cancer-related morbidity and mortality worldwide. The dysregulation of miR-34s, specifically miR-34a, has been implicated in the development and progression of colon carcinoma. A recent study revealed that miR-34a induced immunosuppression in colon carcinoma through the SIRT1/NF-κB/B7-H3/TNF-α axis. The downregulation of miR-34a leads to the increased expression of these immunosuppressive factors, contributing to tumor growth and the evasion of immune surveillance [83]. Furthermore, expressions of miR-34a/b/c genes are frequently silenced in colon carcinoma. They are capable of suppressing tumor formation caused by the loss of the Apc gene, which is commonly associated with colon cancer. Additionally, miR-34s help maintain the homeostasis of intestinal stem cells and secretory cells by regulating various target genes and pathways [87]. In colon carcinoma cells, the expression of miR-34a is often reduced and the restoration of miR-34a expression inhibits colony formation, proliferation, migration, and invasion and induces cell apoptosis through the regulation of SYT1. This suggests that miR-34a could serve as a potential target for colon cancer treatment [84].

Furthermore, TP53 gene polymorphisms are associated with the methylation and expression of miR-34a/b/c in colorectal cancer tissues [113]. This implies that miR-34s could have diagnostic and therapeutic potential in colon cancer management [114]. In addition to miR-34s, lncRNA NONHSAG028908.3 has been identified as a modulator of colorectal cancer growth. LncRNA NONHSAG028908.3 upregulates the expression of aldolase, fructose-bisphosphate A (ALDOA), a protein involved in cancer cell metabolism via acting as a sponge for miR-34. By regulating the expression of ALDOA, lncRNA NONHSAG028908.3 can promote colorectal cancer growth [45]. Understanding the role of miR-34s in colon carcinoma may aid in the development of targeted therapies and diagnostic approaches for improved management and outcomes in patients with colon carcinoma.

### 3.8. The Tumor-Suppressive Effects of miR-34s in Ovarian Cancer

miR-34s hold significant importance in ovarian cancer research. Studies indicate that miR-34a, miR-34b, and miR-34c are notably downregulated in ovarian cancer tissues compared to healthy ones. Furthermore, the expression levels of these miR-34 members serve as independent markers for progression-free survival in ovarian cancer patients [23].

Another intriguing finding is the involvement of exosomal miR-34b in ovarian cancer. It exerts its effects by suppressing ovarian cancer cell proliferation and EMT by targeting Notch2 [76]. The ability of exosomal miRNAs to be transported between cells and influence tumor behavior has made them a potential diagnostic and therapeutic candidate. Furthermore, recent studies have identified potential prognostic markers in ovarian cancer. The overexpression of P2RY14, a G protein-coupled receptor, has been associated with improved overall survival and longer progression-free intervals in ovarian cancer patients. Interestingly, lncRNA LINC00665 acts as a sponge for miR-34c-5p, preventing its interaction with P2RY14. This leads to the maintenance of low P2RY14 expression, which, in turn, correlates with poor prognosis and increased tumor immune infiltration in ovarian carcinoma [80]. The dysregulation of miR-34 family members and their downstream targets in ovarian cancer provides valuable insights into their significant roles in disease progression. This knowledge has the potential to contribute to the development of innovative diagnostic approaches and targeted therapies, ultimately leading to improved management and better outcomes for patients with ovarian cancer.

### 3.9. The Role of miR-34s in Cervical Cancer

MiR-34s exhibit differential effects in cervical cancer, with distinct functions observed for the 5p and 3p strands. Specifically, miR-34a-5p has been shown to inhibit proliferation, migration, and cell invasion by reducing the activity of MMP9 and the protein expression of MAP2 [81]. Similarly, miR-34b-5p and miR-34c-5p demonstrate inhibitory effects on proliferation and migration, while miR-34c-5p specifically suppresses MMP9 activity and MAP2 protein levels [115]. Both miR-34a-3p and miR-34b-3p suppress proliferation and migration, while miR-34c-3p inhibits cell invasion and MMP9 activity and reduces MAP2 protein expression [115]. These findings imply that other regulatory genes may contribute to the varying effects of miR-34s in cervical cancer. Furthermore, specific target genes have been identified for miR-34a and miR-34c in cervical cancer. For instance, miR-34a has been reported to impede growth and migration by targeting CDC25A [116], a key cell cycle regulator. Similarly, miR-34c exerts suppressive effects on metastasis and invasion by targeting Notch 1 [49], a critical signaling pathway involved in cancer progression. These findings highlight the intricate regulatory roles of miR-34s in cervical cancer and provide insights into potential therapeutic targets. Overall, the dysregulation of miR-34s in cervical cancer underscores their significance as potential diagnostic markers and therapeutic targets in the management of this disease. Future studies are warranted to elucidate the complex interplay between miR-34s and their target genes.

### 3.10. The Tumor Suppressive Function of miR-34s in Prostate Cancer

Prostate cancer is the most frequently diagnosed cancer in men, with an estimated 288,300 new cases and 34,700 expected prostate cancer deaths in 2023 [27]. Prostate cancer incidence increased by 3% annually from 2014 to 2019 after two decades of decline, which translates into an additional 99,000 new cases per year [27]. Recent studies have shown the involvement of genetic and epigenetic alterations, as well as microRNAs, including miR-34a, in prostate cancer development [117,118]. Notably, miR-34a has been found to exert tumor-suppressive effects on prostate cancer. In PC3 prostate cancer cells, miR-34a has been shown to inhibit cell growth and migration while promoting G2 cell cycle arrest by targeting the Wnt signaling pathway [88].

The intricate network of lncRNAs has also been implicated in prostate cancer progression by interacting with miR-34a. For instance, lncRNA LINC00662 promotes prostate cancer tumorigenesis by acting as a sponge for miR-34a [119]. Similarly, LINC01006 upregulates the expression of DAAM1, thereby facilitating prostate cancer cell proliferation, migration, and invasion through its interaction with miR-34a-5p [56]. Additionally, lncRNAs DANCR and NEAT1 contribute to docetaxel (DTX) resistance in prostate cancer by sequestering miR-34a-5p [44,63]. This emerging evidence underscores the significance of miR-34s, particularly miR-34a, in prostate cancer biology. Further investigations are warranted to unravel the underlying mechanisms and exploit the therapeutic potential of targeting the miR-34 family in prostate cancer management.

### 3.11. The Role of miR-34s in Osteosarcoma

Osteosarcoma, a highly aggressive bone malignancy primarily affecting children, adolescents, and young adults, poses significant challenges in terms of diagnosis and treatment. In recent years, the role of miR-34s in osteosarcoma has emerged as a promising area of research, shedding light on the intricate molecular mechanisms underlying tumor development and progression. Among the members of the miR-34 family, miR-34a has garnered considerable attention due to its involvement in crucial cellular processes, including cell proliferation, differentiation, migration, and apoptosis. Studies in nude mice have demonstrated the tumor-suppressive effects of miR-34s on osteosarcoma. It has been shown that by regulating the expression of the TGIF2 gene, miR-34s effectively inhibit tumor growth and promote apoptosis, thereby impeding the progression of osteosarcoma [86,120,121].

In addition to its direct impact on tumor cells, miR-34a exerts its influence through intricate signaling pathways within the tumor microenvironment. Notably, the DNMT1/miR-34a/Bcl-2 axis has emerged as a key regulatory pathway in osteosarcoma. Isovitexin, a natural compound, has shown promising potential in disrupting this axis. By targeting DNMT1 and modulating miR-34a expression, isovitexin effectively hampers cancer stemness properties and induces apoptosis in osteosarcoma cells. This finding holds great promise for the development of novel therapeutic strategies that could potentially overcome the treatment challenges posed by osteosarcoma [47,122].

Moreover, the dysregulation of miR-34a has been linked to various target genes and pathways involved in osteosarcoma progression. For instance, miR-34a has been shown to target DUSP1, a protein implicated in cell signaling pathways, as well as the MALAT1/cyclin D1 axis, which plays a crucial role in cell cycle regulation. The dysregulation of these targets contributes to the uncontrolled growth, progression, and metastasis observed in osteosarcoma. In addition, many lncRNAs, such as HCG18 [82], DICER1-AS1 [62], NEAT1 [67], CASC11 [123], MALAT1 [124], and SNHG7 [48], have been identified as miR-34a sponges and can promote osteosarcoma progression by sequestering miR-34a and diminishing its tumor-suppressive effects. Additionally, lncRNA HCG9 has been found to act as a sponge for miR-34b, thereby facilitating osteosarcoma progression through the regulation of RAD51 [81]. LncRNA KCNQ1OT1 sponged miR-34c-5p to promote osteosarcoma growth through ALDOA-enhanced aerobic glycolysis [46]. The intricate network of miR-34s and their target genes underscores the multifaceted role of miR-34a in osteosarcoma pathogenesis. Continued research efforts are essential to fully comprehend the complexity of miR-34s in osteosarcoma and exploit their therapeutic potential in combating this aggressive bone malignancy.

### 3.12. The Role of miR-34s in Leukemia

Leukemia, particularly acute myelogenous leukemia (AML), is a widespread and challenging form of acute leukemia that accounts for approximately 80% of all cases worldwide [125]. In chronic lymphocytic leukemia (CLL), a regulatory circuit has been discovered. The tumor suppressor protein p53 has been found to regulate the expression of miR-34a, which in turn controls the expression of MDM4. This regulatory circuit plays a pivotal role in orchestrating cancer cell apoptosis and the survival of CLL cells [70]. In the context of AML, miR-34a dysregulation has been linked to a circular RNA called circ_POLA2. This circular RNA has been found to impede the maturation of miR-34a, resulting in increased cell proliferation and the aggressiveness of AML [70]. Circular RNA ATAD1, which is upregulated in AML, has been implicated in cancer cell proliferation by downregulating miR-34b via promoter methylation [126]. Furthermore, miR-34c-5p has emerged as a promising target in combating AML stem cells. Its unique mechanism involves inhibiting the shedding of exosomes and inducing cellular senescence through the regulation of RAB27B expression. This finding opens up new avenues for targeting AML stem cells to improve treatment outcomes [127].

These captivating findings underscore the significant role of miR-34s in the intricate landscape of leukemia. By unraveling the dysregulation of miR-34s and their intricate interactions with target genes, invaluable insights into the pathogenesis of leukemia are revealed. Ultimately, this knowledge could pave the way for the development of innovative and targeted therapeutic strategies, providing hope for improved outcomes for leukemia patients. Continued research into the precise mechanisms and crosstalk of miR-34s within the leukemia context holds tremendous promise for advancing our understanding of this complex disease and translating it into tangible clinical benefits.

### 3.13. The Role of miR-34s in Bladder Cancer

Bladder cancer ranks as the 10th most common cancer globally, with high morbidity and mortality rates. In 2023, it is estimated that there will be approximately 19,870 new cases and 4550 expected deaths among women, and 62,420 new cases and 12,160 expected deaths among men in the United States alone [27]. One of the key players in bladder cancer is miR-34a, which has been shown to regulate crucial cellular processes such as cell cycle control, migration, invasion, and autophagy [128]. miR-34a exhibits a dual effect in bladder cancer by upregulating the expression of PTEN, a tumor suppressor gene, while downregulating MPP2, a gene associated with tumor progression [72,129]. Additionally, the silencing of DNMT3B, a DNA methyltransferase, has been found to promote the expression of miR-34a, resulting in the suppression of cancer migration and invasion [130]. In non-muscle-invasive bladder tumors, the expression of miR-34a has emerged as an independent predictor of recurrence, and in conjunction with the EORTC nomogram, the levels of miR-34a can further improve the predictive capabilities of EORTC [131]. This highlights the potential of miR-34a as a valuable biomarker for prognostic assessment and treatment decision-making in bladder cancer. Moreover, a panel of four miRNAs, including miR-34a, miR-182, miR-196a, and miR-124, has been identified as a promising noninvasive biomarker for bladder cancer, exhibiting high diagnostic accuracy with an area under the curve of 0.985 [132].

By elucidating the regulatory roles of miR-34a and other miRNAs in key cellular processes and their implications in diagnostic and prognostic strategies, we can enhance our ability to detect bladder cancer at early stages, predict disease progression, and develop targeted therapeutic interventions.

## 4. Exploring miR-34s in Drug Resistance

Resistance to conventional chemotherapy remains a significant challenge in cancer treatment. Therapy resistance leads to cancer relapse and poor patient outcomes. Numerous studies have demonstrated that altered levels of circulating miR-34s or tumor-specific miR-34 expressions are associated with poor responses to chemotherapy [133,134,135]. In patients with osteosarcoma, serum levels of miR-34a correlated with chemotherapy resistance, metastasis, recurrence, overall survival, and prognosis [28,136]. In colorectal cancers with inactive p53, targeting the miR-34a/LRPPRC/MDR1 axis has shown promise in overcoming resistance to chemotherapeutic agents such as gossypol-acetic acid and 5-fluorouracil (5-FU) [137]. The ability of miR-34a to suppress cancer stem cell renewal has been linked to a reduction in gemcitabine resistance in pancreatic cancer [138]. Furthermore, miR-34a has the ability to reverse multidrug resistance (MDR) to various chemotherapeutic agents including 5-FU, cisplatin (DDP), oxaliplatin, and epirubicin (EPI) in gastric cancer cells [139]. In bladder cancer, exosomal lncRNA LINC00355 derived from cancer-associated fibroblasts promotes resistance to DDP by targeting the miR-34b-5p/ABCB1 axis [140]. miR-34b expression has been reported to enhance paclitaxel chemosensitivity in endometrial cancer cells [141]. miR-34b also suppresses MDR to multiple drugs (paclitaxel, pirarubicin, EPI, hydrochloride, adriamycin, and DDP) in bladder cancer by regulating the expression of CCND2 and P2RY1 [142].

In the case of miR-34c, its role in drug resistance is complex and context-dependent. It has been reported to protect lung cancer cells from paclitaxel-induced apoptosis by regulating the expression of Bmf (Bcl-2-modifying factor) [143]. However, decreased miR-34c expression has also been observed in patients with NSCLC who exhibited a poor response to chemotherapy and increased metastasis. miR-34c overexpression sensitized NSCLC cells to paclitaxel and DDP both in vitro and in vivo [144]. miR-34c also inhibits MDR to paclitaxel and DDP in gastric cancer cells [145]. In ovarian cancer, the miR-34c/SOX9 axis regulates DDP-based drug resistance, and the miR-34c-AREG-EGFR-ERK pathway inhibits amphiregulin-induced cancer stemness and drug resistance [146,147].

Understanding the intricate roles of miR-34s in mediating drug resistance provides valuable insights for developing strategies to overcome treatment limitations and improve patient outcomes. Harnessing the potential of miR-34s as therapeutic targets or predictive biomarkers may lead to the development of personalized treatment approaches and the effective management of drug resistance in various cancer types.

### Overcoming Chemoresistance by Targeting CSCs

Chemoresistance, a common hurdle in cancer treatment, has been attributed to the presence of a small population of CSCs that possess self-renewal and differentiation capabilities [148]. These CSCs are thought to be responsible for tumor initiation, progression, and relapse. Due to its ability to suppress CSC activity, targeting CSCs using miR-34-based therapies has emerged as a promising approach to overcoming chemoresistance and improving treatment outcomes [149]. In taxane-resistant prostate cancer, the codelivery of DTX, a chemotherapy drug, and rub one (RUB), an activator of miR-34a specifically targeting CSCs, showed that the combination therapy effectively suppressed CSC-related markers, reduced the CSC population, and enhanced the therapeutic response compared to DTX alone [148,150]. Furthermore, miR-34 mimics have been shown to sensitize CSCs to chemotherapy alone. A combination of miR-34a mimics and doxorubicin, a commonly used chemotherapy drug used in breast cancer, synergistically inhibited breast cancer CSC properties and significantly reduced tumor growth compared to either treatment alone by sensitizing the CSCs to doxorubicin-induced apoptosis, thereby overcoming chemoresistance [151,152].

In addition to targeting CSCs directly, miR-34-based therapies have been explored for modulating the tumor microenvironment to disrupt CSC-related signaling pathways. For instance, miR-34a mimic delivery using nanocarriers was found to suppress the expression of CSC-associated factors and promote the differentiation of CSCs in hepatocellular carcinoma. This resulted in reduced tumor growth and enhanced chemosensitivity [153]. Moreover, the combination of miR-34 mimics with natural compounds known for their anticancer effects has shown promise in targeting CSCs. For example, the codelivery of miR-34a mimics with curcumin, a natural compound derived from turmeric, demonstrated synergistic effects in inhibiting CSC properties and suppressing tumor growth in colorectal cancer models [154]. Collectively, these studies highlight the potential of miR-34-based therapies in targeting CSCs and overcoming chemoresistance. By specifically inhibiting CSCs and their associated signaling pathways, miR-34 mimics hold promise for improving treatment outcomes and address the challenges posed by CSC-mediated resistance in cancer therapy.

## 5. miR-34s and Cancer Therapy

miR-34s have brought about a paradigm shift in cancer therapy. Their ability to regulate gene expression and control critical processes such as cell proliferation, apoptosis, and metastasis has positioned miR-34s as promising candidates for targeted cancer treatment [28,149,155]. Figure 3 depicts the therapeutic application of miR-34s in cancer therapy and the potential impact of these molecules on combating cancer. In this section, we discuss the potential of miR-34s in cancer therapy and the diverse strategies used to harness their therapeutic potential.

### 5.1. Chemically Synthesized miR-34: Clinical Trials and Beyond

The development of MRX34, a chemically synthesized miR-34a mimic, has opened up new possibilities for clinical trials to investigate the therapeutic potential of miR-34 mimics in various types of cancer. A phase I/II multicenter clinical trial (MRX34, NCT01829971) evaluated the safety and efficacy of MRX34 in patients with primary liver cancer, lymphoma, and lung cancer [156,157,158]. This trial includes 155 participants from seven different cancer types, including primary liver cancer, various solid tumors, and hematopoietic malignancies [156]. The study found that patients treated with MRX34 experienced adverse events, such as fever and fatigue. The MTD was determined to be 110 mg/m^2^ for non-HCC patients and 93 mg/m^2^ for HCC patients [159]. Importantly, MRX34 demonstrated antitumor activity in patients with refractory solid tumors [160]. Additionally, the biodistribution of MRX34 was found to be broad and could be detected in various tissues including the liver, bone marrow, spleen, mammary glands, and lungs [161]. This widespread distribution of MRX34 enables its potential application in the treatment of different cancer types.

Indeed, recent studies have shown promising antitumor activity of MRX34 in various cancer types. In liver tumor xenograft models, the systemic delivery of MRX34 led to a 1000-fold increase in miR-34a levels, resulting in the inhibition of tumor growth [159] and tumor regression in more than one-third of tumor-bearing mice [158]. These encouraging findings have prompted further investigations and sparked interest in exploring the potential of miR-34 mimics in other cancer types. The following are some additional studies to evaluate the efficacy of miR-34 mimics in diverse cancer types and explore their potential as a broad-spectrum therapeutic approach [158].

Lung cancer: Clinical trials are currently underway to evaluate the safety and effectiveness of miR-34 mimics in patients with lung cancer. These trials aim to assess the impact of miR-34a mimic therapy on tumor growth, metastasis, and patient outcomes in different subtypes of lung cancer [162,163]. Additionally, using a preclinical mouse model of NSCLC known as 344SQ, treatment with MRX34 led to a decreased expression of PD-L1 protein, increased infiltration of tumor-fighting CD8^+^ cells, and decreased infiltration of PD1^+^ T-cells, macrophages, and T-regulatory cells, leading to delay in tumor growth [157]. Furthermore, a combination of miR-34a and let-7b by the encapsulated vehicle NOV340 reduced tumor burden and prolonged survival in therapy-resistant NSCLC mouse models [162].Lymphoma: The initial multicenter clinical trial (NCT01829971) included patients with lymphoma and provided evidence for the feasibility and potential benefits of miR-34 mimics therapy in this type of cancer. More studies to investigate the specific effects of miR-34 mimics on different subtypes of lymphoma and patient response rates are currently ongoing [164,165,166].Breast cancer: Clinical trials are being planned or conducted to explore the role of miR-34 mimics in the treatment of breast cancer. These trials aim to evaluate the therapeutic efficacy of miR-34a mimics in combination with standard treatments, such as chemotherapy or targeted therapies [159,167].Prostate cancer: Studies and clinical trials to investigate the potential of miR-34 mimics in the treatment of prostate cancer, including the effects of miR-34a mimics on tumor regression, metastasis, and patient survival rates in prostate cancer patients are ongoing [160,168].

### 5.2. Enhancing miR-34 Efficacy: Chemical Modifications and Nanodelivery Systems

The efficacy of miR-34 in cancer therapy has sparked significant interest in developing new strategies to enhance its therapeutic potential. Chemical modifications and the development of nanodelivery systems have emerged as promising approaches to improving the stability and efficacy of miR-34. These advancements aim to overcome the challenges associated with miRNA delivery, such as poor cellular uptake, rapid degradation, and off-target effects. By incorporating chemical modifications and the development of nanodelivery systems, the goal is to enhance the stability, target specificity, and therapeutic activity of miR-34, paving the way for its effective use in cancer treatment. This section describes the recent advances in enhancing miR-34 efficacy through these innovative strategies.

#### 5.2.1. Chemical Modifications to Enhance Stability and Potency

Chemical modifications are essential for enhancing the stability and potency of miR-34 mimic therapy, with a primary focus on preventing nuclease degradation, reducing off-target effects, and improving efficacy. One commonly employed strategy involves the incorporation of locked nucleic acids (LNAs) into the miR-34 mimic sequence [169]. LNAs are modified RNA nucleotides with enhanced stability and binding affinity by locking the ribose in a ring conformation. The incorporation of LNAs into miR-34 mimics has been demonstrated to significantly improve their resistance to nucleases and increase their intracellular availability, resulting in an enhanced suppression of target genes [170]. Another approach involves the use of 2′-O-methyl modifications, where the 2′-hydroxyl group in the ribose sugar of miR-34 mimics is replaced with a methyl group. This modification enhances their stability and reduces their immunostimulatory effects, thereby improving the pharmacokinetic properties of miR-34 mimics [171]. Furthermore, conjugating miR-34 mimics with cholesterol or other lipid moieties has been tested to improve cellular uptake and intracellular delivery. The hydrophobic nature of these modifications enables their efficient association with cell membranes, facilitating the endocytosis and subsequent release of miR-34 mimics into the cytoplasm [172,173].

#### 5.2.2. Nanodelivery Systems for Enhanced Cellular Uptake

Nanodelivery systems, encompassing liposomes and nanoparticles, have surfaced as highly effective carriers for facilitating the delivery of miR-34 mimics to cancer cells. These systems offer several advantages, including the protection of miR-34 mimics from enzymatic degradation, improved cellular uptake, and controlled release at the target site [174,175]. Liposomes, composed of lipid bilayers, have been extensively used as delivery vehicles for miRNAs [175]. Liposomes can encapsulate miR-34 mimics within their aqueous core or incorporate them into the lipid membrane. Liposomes can shield miR-34 mimics from extracellular degradation, facilitate their internalization into cancer cells through endocytosis, and allow for a controlled release of miR-34 mimics within the cells [159,176]. In addition to liposomes, various types of nanoparticles, such as polymeric nanoparticles and lipid nanoparticles, have been utilized for the delivery of miR-34 mimics [177,178]. These nanoparticles offer advantages such as tunable particle size, stability, and versatile surface modifications, enabling enhanced targeting and cellular uptake. The successful delivery of miR-34 mimics and efficient gene silencing using nanoparticles has been demonstrated [179]. Moreover, incorporating targeting ligands, such as antibodies or peptides, onto the surface of nanodelivery systems can enhance the specific uptake of miR-34 mimics by the cancer cells to improve the therapeutic outcomes [180]. In summary, chemical modifications and nanodelivery systems represent valuable strategies for enhancing the stability, potency, and cellular uptake of miR-34 mimics. These approaches have shown promise in preclinical studies for advancing miR-34-based therapeutics in cancer treatment.

### 5.3. Synergistic Effects of Codelivery of miR-34 Strategies

The codelivery of miR-34 mimics and other therapeutic agents has gained significant attention due to their ability to enhance the efficacy of cancer treatment. By simultaneously targeting multiple pathways involved in tumor growth and progression, the codelivery of therapeutics can create a synergistic effect and amplify overall therapeutic outcomes [150]. Codelivering miR-34 mimics with complementary therapeutics is gaining traction due to its potential to enhance cancer treatment. This approach concurrently targets multiple critical pathways in tumor growth, creating a synergistic effect that boosts treatment outcomes [181,182]. Effective codelivery strategies, including nanodelivery systems, combination therapies, and innovative drug formulations like liposomes and nanoparticles, should align with the specific cancer type and treatment options [183,184]. These multifaceted approaches offer substantial benefits. The selection of specific codelivery strategies should be guided by preclinical studies and clinical trials tailored to the specific cancer type and treatment options [181,182]. Here, we highlight the potential benefits of the codelivery approaches.

#### 5.3.1. Codelivery of miR-34 Mimics with Natural Compounds

Thymoquinone (TQ, a derivative of the Chinese herbal medicine dendrobium) possesses potent antitumor properties, including the inhibition of cell proliferation and induction of apoptosis. When combined with miR-34 mimics, overall better treatment outcomes were observed [152,185]. Another natural compound with anticancer properties is curcumin. The anticancer effects of curcumin are derived from its anti-inflammatory, antioxidant, and proapoptotic activities. The codelivery of miR-34 mimics with curcumin enhances their intracellular uptake and stability, leading to synergistic inhibition of tumor growth, metastasis, and drug resistance [186,187].

#### 5.3.2. Codelivery with Conventional Chemotherapy Drugs

Paclitaxel is a commonly used chemotherapy drug that disrupts microtubule dynamics, leading to cell cycle arrest and apoptosis. The co-delivery of miR-34 mimics with paclitaxel can enhance the cytotoxic effects of paclitaxel by targeting different molecular pathways simultaneously. This approach has the potential to overcome chemoresistance and improves treatment outcomes in various cancer types [188,189]. Similarly, doxorubicin is a potent chemotherapy agent used in the treatment of several cancers. The codelivery of miR-34 mimics with doxorubicin enhances anticancer efficacy by sensitizing cancer cells to the drug’s cytotoxic effects. This combination has shown promise in overcoming drug resistance and improving therapeutic outcomes in preclinical studies [190].

#### 5.3.3. Codelivery with Targeted Therapies

The codelivery of miR-34 mimics with targeted therapies aims to elicit synergistic effects between the targeted therapies and miR-34-based therapy. For example, the codelivery of miR-34 mimics with inhibitors of specific signaling pathways, such as the EGFR inhibitors or PI3K/Akt pathway inhibitors, can result in an enhanced inhibition of tumor growth and improved patient response rates [191]. The combination of miR-34 mimics with complementary therapeutic agents, whether natural compounds or conventional drugs, offers the advantage of targeting multiple pathways simultaneously. This approach can overcome drug resistance, improve treatment efficacy, and potentially reduce side effects associated with high doses of individual drugs.

## 6. Future Perspectives and Conclusions

The emerging roles of miR-34s as critical tumor suppressors in various cancers open up exciting possibilities for their therapeutic applications. Future studies to delineate the complex signaling pathways regulated by miR-34s will not only advance our understanding of their functions but will also identify novel targets for medical intervention. In-depth studies are warranted to elucidate the specific downstream targets of miR-34s and their precise mechanisms of action in different cancer types. This knowledge will enable the development of more specific and personalized miR-34-based therapies [192,193].

While the therapeutic potential of miR-34 mimics is promising, several challenges need to be addressed before its successful translation into clinical practice. One challenge is the efficient and specific delivery of miR-34 mimics to cancer cells. The development of advanced nanodelivery systems to ensure the effective delivery to the tumor site and chemical modifications of miR-34 mimics will further enhance the stability, cellular uptake, and intracellular release of miR-34 mimics. Overcoming issues related to the biodegradation of delivery systems is crucial to maintaining therapeutic efficacy while minimizing potential adverse effects [156,194].

Additionally, the development of miR-34-based therapies faces challenges related to their stability. miR-34 mimics are susceptible to degradation by nucleases in the bloodstream and cellular environment [195]. Therefore, improving the stability of miR-34 mimics through chemical modifications or encapsulation within protective carriers is to achieve therapeutic levels and enhance their therapeutic effects. Another challenge is the cost associated with the production and delivery of miR-34 mimics. The synthesis process of miR-34-based therapeutics and the need for specialized delivery systems contribute to the high cost of miR-34-based therapies. Strategies to optimize and streamline the manufacturing process could help reduce the overall cost, making these therapies more accessible and affordable for patients [103,196].

Moreover, the identification of reliable biomarkers that predict patient responses to miR-34-based therapies is essential. Biomarker discovery efforts are crucial for patient stratification, enabling the selection of individuals who are most likely to benefit from miR-34 mimics. Such personalized treatment approaches would optimize treatment outcomes, minimize potential side effects, and maximize the cost-effectiveness of miR-34-based therapies [197,198].

Considering the complexity of cancer and the dynamic nature of tumor progression, combination therapies that target multiple pathways hold great promise. Combining miR-34 mimics with conventional chemotherapy, targeted therapies, or immunotherapies can potentially overcome drug resistance and improve patient responses. Synergistic effects can be achieved by simultaneously targeting CSCs and bulk tumor cells, disrupting key oncogenic pathways, and reestablishing tumor suppressor functions [193,198].

In conclusion, while miR-34-based therapies show tremendous potential in cancer treatment, several challenges remain to be overcome. Addressing issues related to delivery efficiency, stability, toxicity, and cost-effectiveness will be crucial in realizing the full therapeutic potential of miR-34 mimics. Furthermore, exploring combination therapies and uncovering novel mechanisms of action will contribute to our knowledge and the success of new and effective miR-34-based cancer therapeutics.

## Figures and Tables

**Figure 1 cancers-15-04723-f001:**
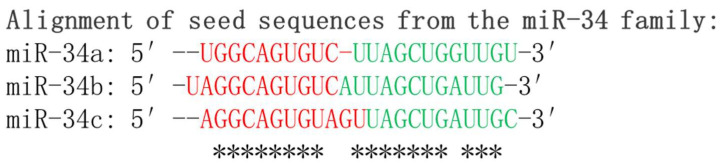
Sequence alignment of mature miR-34 family members: miR-34a, miR-34b, and miR-34c. The alignment showcases the highly conserved regions of these microRNAs, highlighting their functional importance. The seed sequences which are critical for target recognition by the miR-34 family are shown in red. * indicates conserved nucleotides of the seed sequences among family members.

**Figure 2 cancers-15-04723-f002:**
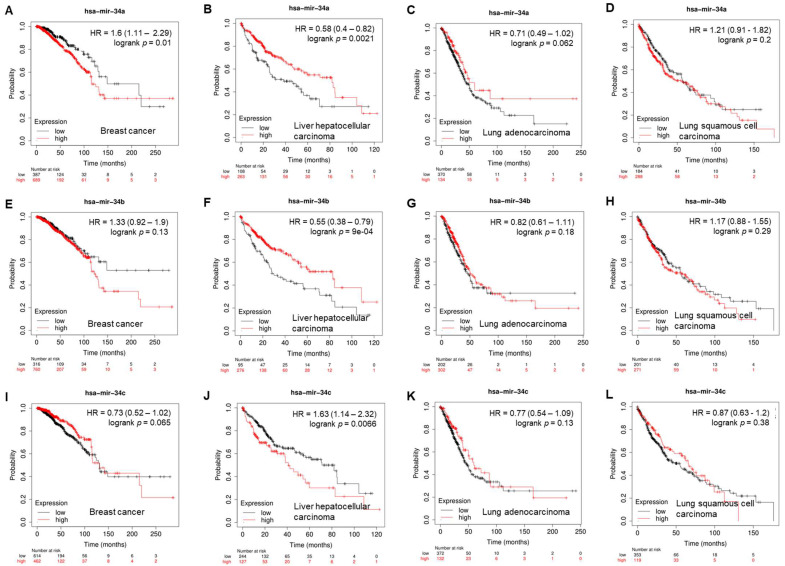
Correlation of overall survival with expression of miR-34a (**A**–**D**), miR-34b (**E**–**H**), and miR-34c (**I**–**L**) in breast cancer, liver hepatocellular carcinoma, lung adenocarcinoma, and lung squamous-cell carcinoma.

**Figure 3 cancers-15-04723-f003:**
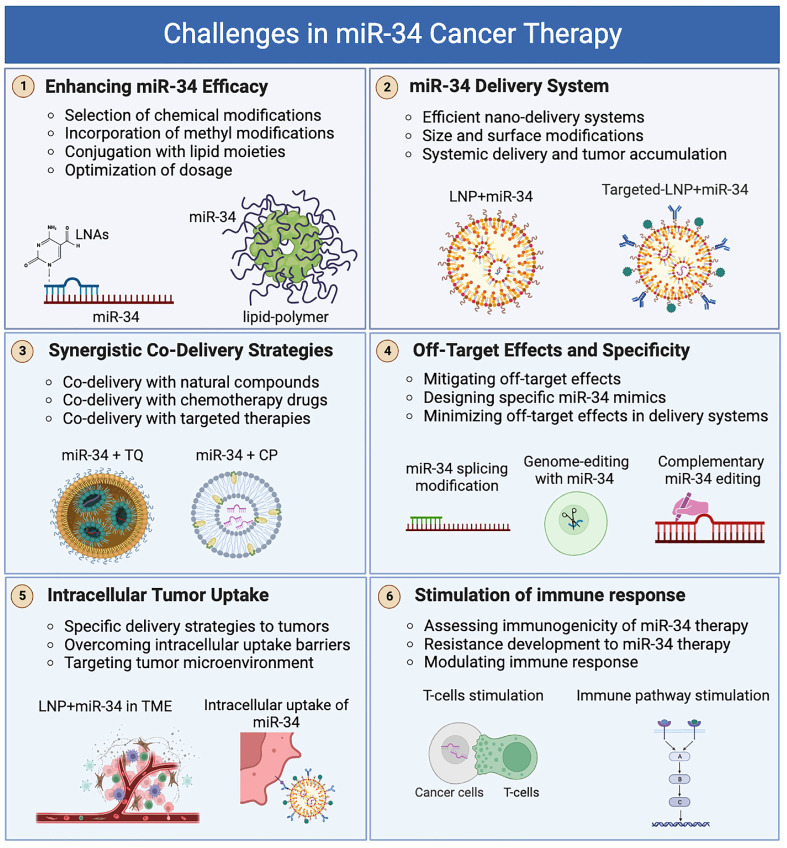
Challenges in miR-34 Cancer Therapy. This schematic illustrates the key obstacles faced in utilizing miR-34 for cancer treatment and the strategies to overcome them. The challenges include improving the efficacy of miR-34 mimics by selecting appropriate modifications, incorporating LNAs and methyl modifications, and conjugating with lipid moieties. Additionally, the figure highlights the importance of optimizing dosage and treatment regimens. Another challenge is the efficient delivery of miR-34, which involves the development of nanodelivery systems with optimized size and surface modifications to achieve systemic delivery and tumor accumulation. The figure also emphasizes the synergistic codelivery strategies of miR-34 with natural compounds, chemotherapy drugs, and targeted therapies to enhance therapeutic outcomes. The design of miR-34 mimics tailored for cancer cells and their delivery systems can minimize off-target effects and improve specificity. The figure also highlights the strategies for specific delivery to tumor sites, overcoming barriers to intracellular uptake, and targeting the tumor microenvironment. The potential of miR-34 therapy in modulating immune responses and the development of resistance to miR-34 therapy are also noted.

**Table 1 cancers-15-04723-t001:** Summary of the current miR-34 targets.

Gene Name	miR-34 Species	Biological Effects	References
*ACSL4/lncRNA NEAT1*	*miR-34a*	Promotion of docetaxel resistance	[44]
*ALDOA/NONHSAG028908.3*	*miR-34a*	Inhibition of cell growth and migration	[45]
*ALDOA/lncRNA KCNQ1OT1*	*miR-34c*	Inhibition of cell growth	[46]
*Bcl-2*	*miR-34a*	Apoptosis, inhibition of cell growth and migration	[47,48]
*CAV1*	*miR-34b, miR-34c*	Inhibition of migration	[20]
*CDC25A*	*miR-34a*	Inhibition of cell growth and migration	[49]
*CDC25C*	*miR-34a*	G2 arrest	[50]
*CCND1*	*miR-34a*	G1 arrest	[51,52]
*CCND2/RP11-805J14.5*	*miR-34a*	Promoted cell cycle progression	[53]
*CCNE2*	*miR-34a, miR-34b, miR-34c*	G1 arrest	[51]
*CDK4*	*miR-34a, miR-34b, miR-34c*	G1 arrest, apoptosis	[54]
*CDK6*	*miR-34a, miR-34b*	G1 arrest	[52]
*CREB*	*miR-34b*	Inhibition of proliferation	[55]
*DAAM1/LINC01006*	*miR-34a*	Inhibition of proliferation, migration, and invasion	[56]
*DLL1*	*miR-34a*	Influence on Notch signaling	[57]
*E2F3*	*miR-34a, miR-34c*	Inhibition of proliferation, senescence	[50]
*E2F5*	*miR-34a*	Transcriptional activation, cell proliferation and migration	[12,58]
*Flotillin-2*	*miR-34a*	Inhibition of proliferation, migratory/invasive activity	[59]
*FOSL1/circCRIM1*	*miR-34c*	Inhibition of proliferation and invasion	[60]
*FOXM1*	*miR-34a*	Inhibition of cell proliferation and cell migration	[61]
*GADD45A/lncRNA DICER1-AS1*	*miR-34a*	Drug resistance	[62]
*JAG1/LncRNA DANCR*	*miR-34a*	Promotion of docetaxel resistance	[63]
*HDAC1/7*	*miR-34a*	Cell survival and therapy resistance	[64]
*HDMX*	*miR-34a*	Increased p53 activity	[51,65]
*HMGA2*	*miR-34a*	Inhibition of proliferation, senescence	[50,66]
*HOXA13/lncNEAT1*	*miR-34a*	Apoptosis	[67]
*MAP2K1*	*miR-34c*	Inhibition of cell migration and invasion	[68]
*MET*	*miR-34a, miR-34b, miR-34c*	G1 arrest, inhibition of invasion and migration	[51,69]
*MDM4*	*miR-34a*	Apoptosis, miR-34a/MDM4/p53 feedback	[70]
*MMP2/MMP9/FNDC3B, MMP2*	*miR-34a*	Inhibition of cell migration and invasion	[71,72]
*MYB*	*miR-34b, miR-34c*	Inhibition of proliferation	[55]
*MYCN/lncRNA LINC01106*	*miR-34a*	Cell viability, invasion, and migration	[73]
*c-MYC*	*miR-34a, miR-34b, miR-34c*	G1 arrest, counteracting cancer stem cell-like properties	[74]
*N-MYC*	*miR-34a*	G1 arrest	[75]
*Notch1*	*miR-34a, miR-34c*	Inhibition of proliferation, metastasis and invasion, apoptosis	[18,48,49]
*Notch2*	*miR-34b*	Inhibition of cell proliferation and EMT	[76]
*PAC1*	*miR-34a*	Apoptosis	[77]
*PD-L1*	*miR-34a*	Inhibiting the growth of TNBC cells	[78,79]
*P2RY14/lncRNA LINC00665*	*miR-34c*	Tumor immune infiltration	[80]
*RAD51/lncRNA HCG9*	*miR-34b*	Proliferation	[81]
*RUNX2/lncRNA HCG18*	*miR-34a*	Inhibition of proliferation, migration, and invasion	[82]
*SFRS2*	*miR-34b, miR-34c*	Influence on miRNA metabolism	[20]
*SIRT1*	*miR-34a*	Increased p53 acetylation and activation (positive feedback loop), immunosuppression	[83]
*SYT1*	*miR-34a*	Apoptosis	[84]
*TBL1XR1*	*miR-34c*	Inhibition of cell proliferation, migration, and invasion	[85]
*TGIF2*	*miR-34s*	Apoptosis	[86]
*WASF1*	*miR-34a, miR-34b, miR-34c*	Suppression of tumor formation	[87]
*TWIST1, ZEB1*	*miR-34a*	Epithelial-to-mesenchymal transition	[18]
*Wnt1*	*miR-34a*	G2 arrest	[88]
*YY1*	*miR-34a*	Apoptosis, inhibition of migration and invasion	[89]

*ALDOA*, fructose-bisphosphate A; *CAV1*, caveolin 1; *CCND1*, cyclin D1; *CCNE2*, cyclin E2; *CREB*, cyclic AMP-responsive element-binding protein; *DAAM1*, disheveled-associated activator of morphogenesis 1; *FOXM1*, forkhead box M1; *GADD45A*, growth arrest and DNA-damage-inducible alpha; *MYB*, *v-myb* myeloblastosis viral oncogene homologue; *SFRS2*, splicing factor arginine/serine-rich 2; *SYT1*, synaptotagmin I; *TGIF2*, transforming growth factor-β-induced factor homeobox 2; *YY1*, Yin Yang-1.

## Data Availability

Not applicable.

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
