# Peer review of "MicroRNA-34 Family in Cancers: Role, Mechanism, and Therapeutic Potential"

_cancers, 2023, doi:10.3390/cancers15194723_

Round 1

Reviewer 1 Report

Overall this review on miR34s is quite exhaustive and properly structured ;

After a brief introduction on miRNA and their roles in different diseases they focuse in a second part on their subject i.e. the miR-34 family ; their biosynthesis and their regulation.

The core part of the review resides in the exhaustive third section describing the roles of this miR-34 family in differents cancer types ; a good review of the litterature on the subject.

Then in the two last sections of the review the authors dissect what are the challenges for the use of these miR family in cancer treatment assays such as methods to better delivery and target, combined therapy,…

This review is a good resume of the current litterature on this particular et miR family and this family is a promising one indeed for new therapy.

Nevertheless the text could benefit from minor modifications and some additions before publishing which will clarify the review for the readers.

An abbreviation list could be useful

line 46 family members

line 54 respectively to what ?

line 59 same respectively to ?

Figure 2 size of police for the number at rik is really small

line 157 which cells ? A transition sentence could be appropriate there

One to three figures could be added to resume by graphical/schematic /tables views for part 2, part 3 and 4 ; with the effects/roles and/or regulations and/or most validated/stricking examples such as metabolism/aldoa and/or autophagy ? ) such as figure 3

line 226 some numbers/statistics (such as for section 3.1)

line 280 cancer statistics ?

line 376 structure of the sentence is incomplete

3.9 as written the differential effects are not clear at all ; as line 427 « on the other hand » … the effects described are similar ; MMP9 and MAP2, proliferation, invasion, migration

5.2.1 and 5.2.2 typo mistake no difference between title and the first sentence.

5.3 one or two sentences could be added on methods/ways to co-deliver

What about toxicity ? Only mentioned in conclusion line 782/808 perhaps a word in the previous section.

line 786 sentence missing a verb

correct with some minor editing required

Author Response

We appreciate the constructive comments to improve our manuscript. We have provided a point-by-point responses and made all the changes in this revised manuscript as requested. 

Reviewer 2 Report

Submitted review article will not be recognized as superior article compared to past review articles.

Since there have been many papers on miR-34-family in the past, it is necessary to clarify the characteristics of this paper.

l  Int J Biochem Cell Biol. 2022 Mar;144:106168. doi: 10.1016/j.biocel.2022.106168. Epub 2022 Jan 24.

l  Cancer Metastasis Rev. 2021 Sep;40(3):925-948. doi: 10.1007/s10555-021-09973-3. Epub 2021 May 6.

l  Oncol Rep. 2019 Nov;42(5):1635-1646. doi: 10.3892/or.2019.7280. Epub 2019 Aug 16.

The lack of figures and tables in this article is boring and unappealing to the reader.

Do you need this diagram (Fig-1)?

The diagram (Fig-2) is blurry.

Is this drawing (Fig-3) your original?

Moderate editing of English language required

Author Response

(The authors gave the same response as above.)

Reviewer 3 Report

In this article by Fu et al. the authors provide an overview of recent progress in research on the role of the miR-34 family in different types of cancer and its potential for cancer therapy. At first, the authors discuss the mechanisms of dysregulation in miR-34 expression in cancer. Next, they focus on signaling pathways and genes that are targeted by miR-34 in various types of cancer. Then, they describe approaches of using miR-34 for medical intervention in cancer treatment.  

This is a nice and comprehensive work that will certainly help a reader to understand the importance of the miR-34 family in cell function and its perspectives for clinical application in cancer treatment.

The review would greatly benefit if the authors added a Table or Figure listing molecular targets of miR-34 reported in the different types of cancer that they extensively described in chapter 3.

English is fine. 

Author Response

(The authors gave the same response as above.)

Round 2

Reviewer 2 Report

The resubmitted paper was revised according to the reviewer’s comments.

Author Response

To be consistent with the rest of the text, the following changes had been made. 

・In Fig. 2, miR34a, miR34b, miR34c. In page 4, line 156 -miR34s. (All should be miR-34x)

Response: In Fig. 2, changed to miR-34a, miR-34b, and miR-34c. In page 4, line 156, changed to miR-34s.

・In page 22, lines 915 and 928, Co-Delivery and Co-delivery.

Response: In page 22, changed to co-delivery.